# Latent Video Dataset Distillation

Ning Li, Antai Andy Liu, Jingran Zhang, Justin Cui
University of California, Los Angeles
Los Angeles, CA 90095

`{ningli23, antailiu, zhangjingran, justincui}@ucla.edu`

## Abstract

*Dataset distillation has demonstrated remarkable effectiveness in high-compression scenarios for image datasets. While video datasets inherently contain greater redundancy, existing video dataset distillation methods primarily focus on compression in the pixel space, overlooking advances in the latent space that have been widely adopted in modern text-to-image and text-to-video models. In this work, we bridge this gap by introducing a novel video dataset distillation approach that operates in the latent space using a state-of-the-art variational encoder. Furthermore, we employ a diversity-aware data selection strategy to select both representative and diverse samples. Additionally, we introduce a simple, training-free method to further compress the distilled latent dataset. By combining these techniques, our approach achieves a new state-of-the-art performance in dataset distillation, outperforming prior methods on all datasets, e.g. on HMDB51 IPC 1, we achieve a 2.6% performance increase; on MiniUCF IPC 5, we achieve a 7.8% performance increase. Our code is available at* [https://github.com/Ning9319/Latent_Video_Dataset_Distillation](https://github.com/Ning9319/Latent_Video_Dataset_Distillation)

## 1. Introduction

Dataset distillation has emerged as a pivotal technique for compressing large-scale datasets into computationally efficient representations that retain their essential characteristics [38]. While this technique has seen remarkable success in compressing image datasets [4, 5, 22, 27, 36, 44], applications onto video datasets remain an underexplored challenge. Videos inherently possess temporal redundancy, as characterized by consecutive frames often sharing substantial similarity, presenting the potential for optimization via dataset distillation.

Existing video distillation methods predominantly focus on pixel-space compression. VDSD [39] addresses the temporal information redundancy by disentangling static and dynamic information. Method IDTD [48] tackles the within-sample and inter-sample redundancies by leveraging a joint-optimization framework. However, these frameworks overlook the potential of latent-space compressions, which have proven transformative in generative models for images and videos [34, 47]. Modern variational autoencoders (VAEs) [29, 40] offer a pathway to address this gap by encoding videos into compact, disentangled representations in latent space.

In this work, we improve video distillation by operating entirely in the latent space of a VAE. Our framework distills videos into low-dimensional latent codes, leveraging the VAE's ability to model temporal dynamics [47]. Unlike previous methods, our approach encodes entire video sequences into coherent latent trajectories to model temporal dynamics through its hierarchical architecture. We compress the VAE itself through post-training quantization, largely reducing the model size, while retaining accuracy [5]. After distillation, we apply Diversity-Aware Data Selection using Determinantal Point Processes (DPPs) [17] to select both representative and diverse instances. Unlike clustering-based or random sampling methods, DPPs inherently favor diversity by selecting samples that are well-spread in the latent space, reducing redundancy while ensuring comprehensive feature coverage [26]. This leads to a more informative distilled dataset that enhances downstream model generalization.

Our method further introduces a training-free latent compression strategy, which uses high-order singular value decomposition (HOSVD) to decompose spatiotemporal features into orthogonal subspaces [39]. This isolates dominant motion patterns and spatial structures, enabling further compression while preserving essential dynamics [34]. By factorizing latent tensors, we dynamically adjust the rank of the distilled representations, allowing denser instance packing under fixed storage limits. Experiments on the Mini-UCF dataset demonstrate that our method outperforms prior pixel-space approaches by 11.5% in absolute accuracy for IPC 1 and 7.8% for IPC 5.

Overall, our contributions are:

1. We propose the first video dataset distillation framework

operating in the latent space, leveraging a state-of-the-art VAE to efficiently encode spatiotemporal dynamics.

2. We address the challenge of sparsity in the video latent space by integrating Diversity-Aware Data Selection using DPPs and High-Order Singular Value Decomposition (HOSVD) for structured compression.

3. Our method generalizes to both small-scale and large-scale video datasets, achieving a new state-of-the-art performance on all settings compared to existing methods.

## 2. Related Work

**Coreset Selection** Coreset selection aims to identify a small but representative subset of data that preserves the essential properties of the full dataset, reducing computational complexity while maintaining model performance. One of the foundational approaches utilizes k-center clustering [30] to formulate coreset selection as a geometric covering problem, where a subset of data points is chosen to maximize the minimum distance to previously selected points. By iteratively selecting the most distant samples in feature space, this method ensures that the coreset provides broad coverage of the dataset's distribution, making it a strong candidate for reducing redundancy in large-scale datasets. Herding methods [40] take an optimization-driven approach to coreset selection by sequentially choosing samples that best approximate the mean feature representation of the dataset. Probabilistic techniques leverage Bayesian inference [24] and divergence minimization [33] to construct coresets that balance diversity and statistical representativeness. Influence-based selection methods [41] instead focus on quantifying the contribution of individual samples to generalization performance, retaining only the most impactful data points.

**Image Dataset Distillation** Dataset distillation [38] has emerged as a powerful paradigm for compressing large-scale image datasets while preserving downstream task performance. Early gradient-based methods like Dataset Distillation (DD) [38] optimized synthetic images by matching gradients between training trajectories on original and distilled datasets. Later works introduced dataset condensation with gradient matching [46]. Further, Meta-learning frameworks Like Matching Training Trajectories (MTT) [3] and Kernel Inducing Points (KIP) [28] advances performance by distilling datasets through bi-level optimization over neural architectures. Dataset condensation with Distribution Matching (DM) [45] synthesizes condensed datasets by aligning feature distributions between original and synthetic data across various embedding spaces.

Representative Matching for Dataset Condensation (DREAM) [21] improved sample efficiency by selecting representative instances that retained the most informative patterns from the original dataset, reducing redundancy in synthetic samples. Generative modeling techniques have also been explored, with Distilling Datasets into Generative Models (DiM) [37] encoding datasets into latent generative spaces, allowing for smooth interpolation and novel sample generation. Similarly, Hybrid Generative-Discriminative Dataset Distillation (GDD) [19] balanced global structural coherence with fine-grained detail preservation by combining adversarial generative models with traditional distillation objectives. However, temporal redundancy and frame sampling complexities, as noted in [11, 20], highlight the unique difficulties of extending image-focused distillation to video datasets.

**Video Dataset Distillation** While dataset distillation has achieved significant success in static image datasets, direct application to videos presents unique challenges due to temporal redundancy and the need for efficient frame selection [34]. Recent attempts to address video dataset distillation have primarily focused on pixel-space compression. Video Distillation via Static-Dynamic Disentanglement (VDSD) [39] tackles temporal redundancies between frames by separating static and dynamic components. VDSD partitions videos into smaller segments and employs learnable dynamic memory block that captures and synthesizes motion patterns, improving information retention while reducing redundancy. IDTD [48] addresses the challenges of within-sample redundancy and inter-sample redundancy simultaneously. IDTD employs an architecture represented by a shared feature pool alongside multiple feature selectors to selectively condense video sequences while ensuring sufficient motion diversity. To retain the temporal information of synthesized videos, IDTD introduces a Temporal Fusor that integrates diverse features into the temporal dimension.

**Text-to-Video Models and Their Role in Latent Space Learning** Latent-space representations have become a cornerstone of modern video modeling, offering structured compression while maintaining high-level semantic integrity [34, 47]. Variational autoencoders provide to enable efficient storage and reconstruction [13]. Extending this concept, hierarchical autoregressive latent prediction [31] introduces an autoregressive component that improves temporal coherence, leading to high-fidelity video reconstructions. Further enhancing latent representations, latent video diffusion transformers [23] incorporate diffusion-based priors to refine video quality while minimizing storage demands.

Building upon these latent space techniques, recent text-to-video models have demonstrated their capability to generate high-resolution video content from textual descriptions. These methods employ a combination of transformer-based encoders and diffusion models to synthesize realistic video sequences. Imagen Video leverages cascaded video diffusion models to progressively upsample spatial and temporal dimensions, ensuring high-quality output [9]. Mean-

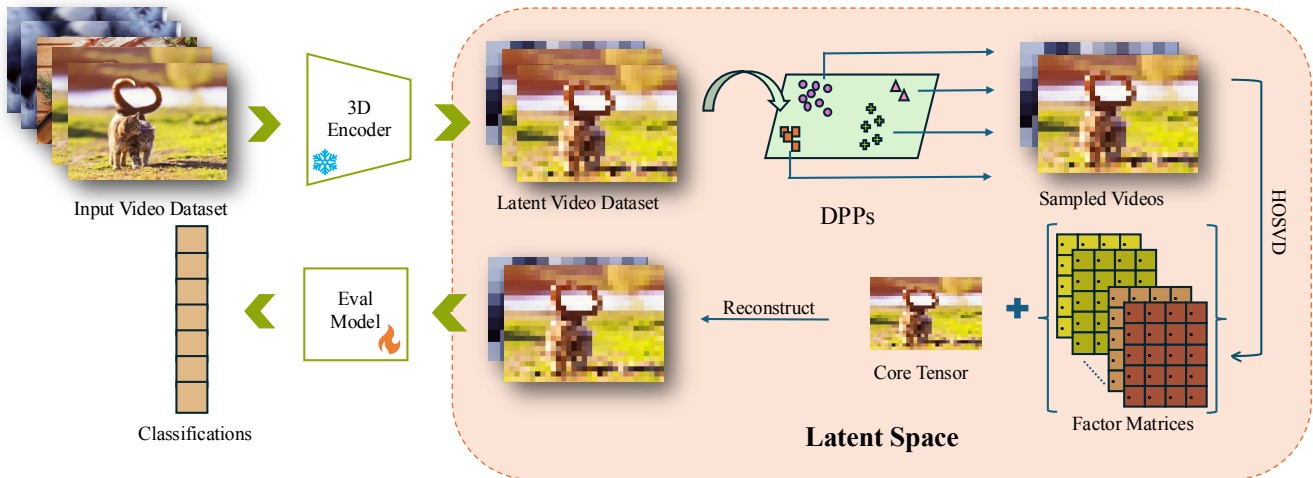

Figure 1. Our training-free latent video distillation pipeline. The entire video dataset is encoded into latent space with a VAE. We further employ the DPPs to select both representative and diverse samples, followed by latent space compression with HOSVD for efficient storage.

while, zero-shot generation approaches utilize decoder-only transformer architectures to process multimodal inputs, such as text and images, without requiring explicit video-text training data [15]. Hybrid techniques combining pixel-space and latent-space diffusion modeling further enhance computational efficiency while maintaining visual fidelity by leveraging learned latent representations during synthesis [43]. These advancements in latent space learning not only improve video compression but also drive the development of scalable and high-quality text-driven video generation.

## 3. Methodology

In this section, we first introduce the variational autoencoder (VAE) used to encode video sequences into a compact latent space. We then discuss our Diversity-Aware Data Selection method. Next, we present our training-free latent space compression approach using High-Order Singular Value Decomposition (HOSVD). Finally, we describe our two-stage dynamic quantization strategy. The entire pipeline of our framework is shown in Fig. 1.

### 3.1. Preliminary

**Problem Definition** In video dataset distillation, given a large dataset $\mathcal{T} = \{(x_i, y_i)\}_{i=1}^{|\mathcal{T}|}$ consisting of video samples $x_i$ and their corresponding class labels $y_i$, the objective is to construct a significantly smaller distilled dataset $\mathcal{S} = \{\tilde{x}_i, \tilde{y}_i\}_{i=1}^{|\mathcal{S}|}$, where $|\mathcal{S}| \ll |\mathcal{T}|$. The distilled dataset is expected to achieve comparable performance to the original dataset on action classification tasks while significantly reducing storage and computational requirements.

**Latent image distillation** has emerged as an effective alternative to traditional dataset distillation methods. In-

stead of distilling datasets at the pixel level, latent distillation leverages pre-trained autoencoders or generative models to encode images into a compact latent space. Latent Dataset Distillation with Diffusion Models [25], have demonstrated that distilling image datasets in the latent space of a pre-trained diffusion model improves generalization and enables higher compression ratios while maintaining image quality. Similarly, Dataset Distillation in Latent Space [6] adapts conventional distillation methods like Gradient Matching, Feature Matching, and Parameter Matching to the latent space, significantly reducing computational overhead while achieving competitive performance. Different from these methods, we extend latent space distillation to video datasets by encoding both spatial and temporal information into the latent space.

### 3.2. Variational Autoencoder

Variational Autoencoders (VAEs) are a class of generative models that encode input data into a compact latent space while maintaining the ability to reconstruct the original data [13]. Unlike traditional autoencoders, VAEs enforce a probabilistic structure on the latent space by learning a distribution rather than a fixed mapping. This allows for better generalization and meaningful latent representations.

A VAE consists of an encoder and a decoder. The encoder maps the input $x$ to a latent distribution $q_\phi(z|x)$, where $z$ is the latent variable. Instead of producing a deterministic latent representation, the encoder outputs the mean and variance of a Gaussian distribution, from which samples are drawn. This ensures that the latent space remains continuous, facilitating smooth interpolation between data points [14]. The decoder then reconstructs the input $x$ from a sampled latent variable $z$, following the learned distribu-

tion $p_\theta(x|z)$.

To ensure a structured latent space, VAEs introduce a regularization term that aligns the learned distribution with a prior distribution, typically a standard normal distribution $p(z) = \mathcal{N}(0, 1)$. This prevents the model from collapsing into a purely memorized representation of the data, encouraging better generalization.

The training objective of a VAE is to maximize the Evidence Lower Bound (ELBO) [18], which consists of two terms: reconstruction loss and Kullback-Leibler (KL) divergence regularization [13]. The reconstruction loss ensures that the decoded output remains similar to the original input, while the KL divergence forces the learned latent distribution to be close to the prior, preventing overfitting and promoting smoothness in the latent space. The overall loss function is formulated as follows.

$$\mathcal{L}_{\text{VAE}} = \mathbb{E}_{q_\phi(z|x)}[-\log p_\theta(x|z)] + \beta \cdot D_{\text{KL}}(q_\phi(z|x) \parallel p_\theta(z)) \tag{1}$$

### 3.3. Diversity-Aware Data Selection

After encoding the entire video dataset into the latent space using a state-of-the-art VAE, an effective data selection strategy is crucial to maximize the diversity and representativeness of the distilled dataset. To this end, we employ Diversity-Aware Data Selection using Determinantal Point Processes (DPPs) [17], a principled probabilistic framework that promotes diversity by favoring sets of samples that are well-spread in the latent space.

DPPs [17] provide a natural mechanism for selecting a subset of latent embeddings that balance coverage and informativeness while reducing redundancy. Given the encoded latent representations of the dataset, we construct a similarity kernel matrix $L$, where each entry $L_{ij}$ quantifies the pairwise similarity between latent samples $z_i$ and $z_j$. The selection process then involves sampling from a determinantal distribution parameterized by $L$, ensuring that the chosen subset is both diverse and representative of the full latent dataset. We define a kernel matrix $L$ using the following function:

$$L_{ij} = \exp(-\frac{\parallel z_i - z_j \parallel^2}{2\sigma^2}) \tag{2}$$

Then subset $S$ is sampled according to:

$$P(S) = \frac{\det(L_S)}{\det(L + I)} \tag{3}$$

here $L_S$ is the submatrix of $L$ that corresponds to the rows and columns indexed by $S$. The denominator $\det(L + I)$ serves as a normalization factor, ensuring that the probabilities across all possible subsets sum to 1. This normalization

stabilizes the sampling process by incorporating an identity matrix $I$, which prevents numerical instability in cases where $L$ is near-singular.

Our approach is motivated by the observation that naive random sampling or traditional clustering-based selection strategies [12] tend to underperform in high-dimensional latent spaces [7], where redundancy is prevalent. By leveraging DPPs, we effectively capture a more comprehensive distribution of video features, thereby improving the quality of the distilled dataset. Furthermore, the computational efficiency of DPPs sampling allows us to scale our selection process to large datasets without significant overhead.

Applying DPPs in the latent space instead of the pixel space offers several key advantages. First, latent representations encode high-level semantic features, making it possible to directly select samples that preserve meaningful variations in motion and structure, rather than relying on pixel-wise differences that may be redundant or noisy. Second, the latent space is significantly more compact and disentangled, allowing DPPs to operate more effectively with reduced computational complexity compared to pixel-space selection [39], which often involves large-scale feature extraction. Finally, in the latent space, similarity measures are inherently more structured, which makes DPPs better suited for ensuring diverse and representative selections that generalize well to downstream tasks.

### 3.4. Training-free Latent Space Compression

While our Diversity-Aware Data Selection effectively distills a compact subset of the latent dataset, we observe that the selected latent representations remain sparse, leading to inefficiencies in storage and downstream processing.

Singular Value Decomposition (SVD) is a fundamental matrix factorization technique widely used in dimensionality reduction, data compression, and noise filtering. Given a matrix $X \in \mathbb{R}^{m \times n}$, SVD decomposes it into three components:

$$X = U\Sigma V^T \tag{4}$$

where U is an orthogonal matrix whose columns represent the left singular vectors, $\Sigma$ is a diagonal matrix containing the singular values that indicate the importance of each corresponding singular vector, and $V$ is an orthogonal matrix whose columns represent the right singular vectors. A key property of SVD is that truncating the smaller singular values allows for an effective low-rank approximation of the original matrix, reducing storage requirements while preserving essential information. This property makes SVD particularly useful in data compression and feature selection.

However, when applied to higher-dimensional data, such as video representations in latent space, SVD requires flattening the tensor into a 2D matrix, which disrupts spa-

| Dataset | | MiniUCF | | HMDB51 | | Kinetics-400 | | SSv2 | |
|---|---|---|---|---|---|---|---|---|---|
| IPC | | 1 | 5 | 1 | 5 | 1 | 5 | 1 | 5 |
| Full Dataset | | $57.2 \pm 0.1$ | | $28.6 \pm 0.7$ | | $34.6 \pm 0.5$ | | $29.0 \pm 0.6$ | |
| Coreset Selection | Random | $9.9 \pm 0.8$ | $22.9 \pm 1.1$ | $4.6 \pm 0.5$ | $6.6 \pm 0.7$ | $3.0 \pm 0.1$ | $5.6 \pm 0.0$ | $3.2 \pm 0.1$ | $3.7 \pm 0.0$ |
| | Herding [40] | $12.7 \pm 1.6$ | $25.8 \pm 0.3$ | $3.8 \pm 0.2$ | $8.5 \pm 0.4$ | $4.3 \pm 0.3$ | $8.0 \pm 0.1$ | $4.6 \pm 0.3$ | $6.8 \pm 0.2$ |
| | K-Center [30] | $11.5 \pm 0.7$ | $23.0 \pm 1.3$ | $3.1 \pm 0.1$ | $5.2 \pm 0.3$ | $3.9 \pm 0.2$ | $5.9 \pm 0.4$ | $3.8 \pm 0.5$ | $4.0 \pm 0.1$ |
| Dataset Distillation | DM [45] | $15.3 \pm 1.1$ | $25.7 \pm 0.2$ | $6.1 \pm 0.2$ | $8.0 \pm 0.2$ | $6.3 \pm 0.0$ | $9.1 \pm 0.9$ | $4.1 \pm 0.4$ | $4.5 \pm 0.3$ |
| | MTT [3] | $19.0 \pm 0.1$ | $28.4 \pm 0.7$ | $6.6 \pm 0.5$ | $8.4 \pm 0.6$ | $3.8 \pm 0.2$ | $9.1 \pm 0.3$ | $3.9 \pm 0.2$ | $6.5 \pm 0.2$ |
| | FRePo [49] | $20.3 \pm 0.5$ | $30.2 \pm 1.7$ | $7.2 \pm 0.8$ | $9.6 \pm 0.7$ | $-$ | $-$ | $-$ | $-$ |
| | DM+VDSD [39] | $17.5 \pm 0.1$ | $27.2 \pm 0.4$ | $6.0 \pm 0.4$ | $8.2 \pm 0.1$ | $6.3 \pm 0.2$ | $7.0 \pm 0.1$ | $4.3 \pm 0.3$ | $4.0 \pm 0.3$ |
| | MTT+VDSD [39] | $23.3 \pm 0.6$ | $28.3 \pm 0.0$ | $6.5 \pm 0.1$ | $8.9 \pm 0.6$ | $6.3 \pm 0.1$ | $11.5 \pm 0.5$ | $5.7 \pm 0.2$ | $8.4 \pm 0.1$ |
| | FRePo+VDSD [39] | $22.0 \pm 1.0$ | $31.2 \pm 0.7$ | $8.6 \pm 0.5$ | $10.3 \pm 0.6$ | $-$ | $-$ | $-$ | $-$ |
| | IDTD [48] | $22.5 \pm 0.1$ | $33.3 \pm 0.5$ | $9.5 \pm 0.3$ | $16.2 \pm 0.9$ | $6.1 \pm 0.1$ | $12.1 \pm 0.2$ | $-$ | $-$ |
| | **Ours** | $\mathbf{34.8 \pm 0.5}$ | $\mathbf{41.1 \pm 0.6}$ | $\mathbf{12.1 \pm 0.3}$ | $\mathbf{17.6 \pm 0.4}$ | $\mathbf{9.0 \pm 0.1}$ | $\mathbf{13.8 \pm 0.1}$ | $\mathbf{6.9 \pm 0.6}$ | $\mathbf{10.5 \pm 0.4}$ |

Table 1. Performance comparison between our method and existing baselines on both small-scale and large-scale datasets. Follow previous works, we report Top-1 test accuracies (%) for small-scale datasets and Top-5 test accuracies (%) for large-scale datasets.

tial and temporal correlations. This limitation motivates our adoption of High-Order Singular Value Decomposition (HOSVD), which extends SVD to multi-dimensional tensors while preserving their inherent structure.

HOSVD is a tensor decomposition technique that generalizes traditional SVD to higher-dimensional data. By treating the selected latent embeddings as a structured tensor rather than independent vectors, we exploit multi-modal correlations across feature dimensions to achieve more efficient compression. Specifically, given a set of selected latent embeddings $Z \in R^{d_1 \times d_2 \times \cdots \times d_n}$, we decompose it into a core tensor $\mathcal{G}$ and a set of orthonormal factor matrices $U_i$, such that

$$Z = \mathcal{G} \times_1 U_1 \times_2 U_2 \times \cdots \times_n U_n \quad (5)$$

where $\times_i$ denotes the mode-$i$ tensor-matrix product. By truncating the singular values in each mode with a rank compression ratio, we discard low-energy components while preserving the most informative structures in the latent space.

A key advantage of HOSVD over traditional SVD is its ability to retain the original tensor structure, rather than requiring flattening into a 2D matrix. More importantly, truncating the singular values in the temporal mode directly reduces temporal redundancy, ensuring that only the most representative motion patterns are retained. This enables more efficient storage and reconstruction, while minimizing the loss of critical temporal information.

Unlike conventional post-hoc compression techniques that require fine-tuning or retraining, HOSVD operates in a completely training-free manner, making it highly efficient and scalable. Furthermore, our empirical analysis shows that applying HOSVD after DPPs-based selection leads to a substantial reduction in storage and computational requirements while maintaining near-optimal performance in downstream tasks.

By integrating HOSVD into our dataset distillation pipeline, we achieve an additional compression gain with minimal loss of information, further pushing the boundaries of efficiency in video dataset distillation.

### 3.5. VAE Quantization

To further improve storage efficiency, we apply a two-stage quantization process to the 3D-VAE [47], combining dynamic quantization for fully connected layers and mixed-precision optimization for all other layers.

The first stage involves dynamic quantization, where all fully connected layers are reduced from 32-bit floating-point to 8-bit integer representations. Dynamic quantization works by scaling activations and weights dynamically during inference. Formally, given an activation $x$ and weight matrix $W$, the quantized representation is computed as:

$$W_q = \text{round}\left(\frac{W}{s_W}\right) + z_W, \quad x_q = \text{round}\left(\frac{x}{s_x}\right) + z_x \quad (6)$$

where $s_W$ and $s_x$ are learned scaling factors, and $z_W$ and $z_x$ are zero points for weight and activation quantization, respectively. The dynamically scaled operation ensures that numerical stability is preserved while reducing the model size. This quantization is applied to all fully connected layers in the encoder and decoder, allowing for efficient memory compression without requiring retraining.

Unlike convolutional layers, fully connected layers primarily perform matrix multiplications, which exhibit high redundancy and are well-suited for integer quantization (INT8). Quantizing these layers from FP32 to INT8 significantly reduces memory consumption and improves computational efficiency while maintaining inference stability [10]. Since fully connected layers do not require the high dynamic range of floating-point precision, INT8 quantization achieves optimal storage and performance benefits.

In the second stage, we employ mixed-precision optimization, where all remaining convolutional and batch normalization layers undergo reduced-precision floating-point compression, scaling them from FP32 to FP16. Unlike integer quantization, FP16 maintains a wider dynamic range,

preventing significant loss of information in convolutional layers, which are more sensitive to precision reduction [42].

This hybrid quantization approach balances storage efficiency and numerical precision, ensuring that the 3D-VAE remains compact while preserving its ability to model spatiotemporal dependencies in video sequences. While applying post-training dynamic quantization on CV-VAE[47], we achieve a more than 2.6× compression ratio while maintaining high reconstruction fidelity.

## 4. Experiments

### 4.1. Datasets and Metrics

Following previous works VDSD [39] and IDTD [48], we evaluate our proposed video dataset distillation approach on both small-scale and large-scale benchmark datasets. For small-scale datasets, we utilize MiniUCF [39] and HMDB51 [16], while for large-scale datasets, we conduct experiments on Kinetics [2] and Something-Something V2 (SSv2) [8]. MiniUCF is a miniaturized version of UCF101 [32], consisting of the 50 most common action classes selected from the original UCF101 dataset. HMDB51 is a widely used human action recognition dataset containing 6,849 video clips across 51 action categories. Kinetics is a large-scale video action recognition dataset, available in different versions covering 400, 600, or 700 human action classes. SSv2 is a motion-centric video dataset comprising 174 action categories.

### 4.2. Baselines

Based on previous work, we include the following baseline: (1) coreset selection methods such as random selection, Herding [40], and K-Center [30], and (2) dataset distillation methods including DM [45], MTT [3], FRePo [49], VDSD [39], and IDTD [48]. DM [45] ensures that the models trained on the distilled dataset produce gradient updates similar to those trained on the full dataset. MTT [3] improves distillation by aligning model parameter trajectories between the synthetic and original datasets. FRePo [49] focuses on generating compact datasets that allow pre-trained models to quickly recover their original performance with minimal training. VDSD [39] introduces a static-dynamic disentanglement approach for video dataset distillation. IDTD [48] enhances video dataset distillation by increasing feature diversity across samples while densifying temporal information within instances.

### 4.3. Implementation Details

**Dataset Details** For small-scale datasets, MiniUCF and HMDB51, we follow the settings from previous work [39, 48], where videos are dynamically sampled to 16 frames with a sampling interval of 4. Each sampled frame is then cropped and resized to 112×112 resolution. We

adopt the same settings as prior work [39, 48] for Kinetics-400, each video is sampled to 8 frames and resized to 64×64, maintaining a compact representation suitable for large-scale dataset distillation. In Something-Something V2 (SSv2), which is relatively smaller among the two large-scale datasets, we sample 16 frames per video and resize them to 112×112, demonstrating the scalability of our method across datasets of varying sizes.

**Evaluation Network** Following the previous works, we use a 3D convolutional network, C3D [35] as the evaluation network. C3D [35] is trained on the distilled datasets generated by our method. Similar to previous works, we assess the performance of our distilled datasets by measuring the top-1 accuracy on small-scale datasets and top-5 accuracy on large-scale datasets.

**Fair Comparison** Throughout our experiments, we rigorously ensure that the total storage space occupied by the quantized VAE model and the decomposed matrices remain within the constraints of the corresponding Instance Per Class (IPC) budget. Specifically, on SSv2, our method utilizes no more than 68% of the storage space allocated to the baseline methods DM and MTT, guaranteeing a fair and consistent comparison. A comprehensive analysis of fair comparisons across all four video datasets is provided in the supplementary material.

### 4.4. Experimental Results

In Tab. 1, we present the performance of our method across MiniUCF [39], HMDB51 [16], Kinetics-400 [2], and SSv2 [8] under both IPC 1 and IPC 5 settings.

On MiniUCF, our approach outperforms the best baseline (IDTD) by 12.3% under IPC 1, achieving 34.8% accuracy compared to 22.5%, and by 7.8% under IPC 5, reaching 41.1% accuracy. Similarly, on HMDB51, our method achieves 12.1% accuracy under IPC 1, surpassing the strongest baseline by 2.6%, while under IPC 5, it reaches 17.6%, a 1.4% improvement. These results highlight the effectiveness of our latent-space distillation framework, which provides superior compression efficiency and classification performance compared to pixel-space-based approaches. The consistent performance gains across both IPC settings demonstrate the robustness of our method in preserving essential video representations while achieving high compression efficiency.

Furthermore, the results in Kinetics-400 and SSv2 reinforce our findings, as our approach consistently outperforms all baselines. Improvements in low-IPC regimes (IPC 1) suggest that our training-free latent compression and diversity-aware data selection are particularly effective when dealing with extreme data reduction. Our method achieves 9.0% accuracy on Kinetics-400 IPC 1, outperforming the strongest baseline (IDTD) by 2.9%, and 6.9% accuracy on SSv2 IPC 1, surpassing VDSD by 2.2%. The

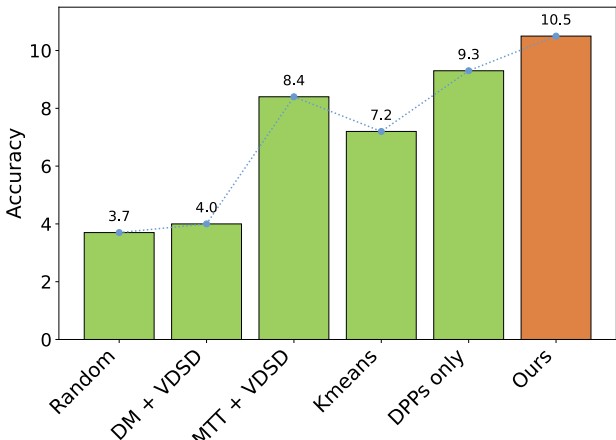

Figure 2. Comparison between different dataset distillation methods and data sampling methods on the SSv2 when IPC is 5.

trend continues in IPC 5, where our model achieves 13.8% on Kinetics-400 and 10.5% on SSv2, both establishing new state-of-the-art results in video dataset distillation.

### 4.5. Ablation Study

In this section, we systematically analyze the key components of our method to understand their contributions to overall performance. We evaluate cross-architecture generalization, various sampling methods, different rank compression ratios in HOSVD, and different latent space compression techniques.

**Cross Architecture Generalization** To further evaluate the generalization capability of our method, we conduct experiments on cross-architecture generalization, as presented in Tab. 2. The results demonstrate that datasets distilled using our method consistently achieve superior performance across different evaluation models—ConvNet3D, CNN+GRU, and CNN+LSTM—compared to previous state-of-the-art methods.

Our approach achieves 34.8% accuracy with ConvNet3D, significantly surpassing all baselines, including MTT+VDSD (23.3%) and DM+VDSD (17.5%). Notably, our method also outperforms all baselines when evaluated on recurrent-based architectures (CNN+GRU and CNN+LSTM), obtaining 19.9% and 18.3% accuracy, respectively. This highlights the robustness of our distilled dataset in preserving spatiotemporal coherence, which is crucial for models that leverage sequential dependencies.

**Sampling Methods** We evaluate the impact of different sampling strategies on dataset distillation, comparing our Diversity-Aware Data Selection using Determinantal Point Processes (DPPs) against random sampling, Kmeans clustering [12], and prior dataset distillation methods (DM + VDSD, MTT + VDSD). As shown in Fig. 2, our method achieves the highest performance, demonstrating the effec-

tiveness of DPPs-based selection in video dataset distillation.

Among sampling strategies, DPPs-only selection outperforms Kmeans and random sampling, indicating that DPPs promote a more diverse and representative subset of the latent space. Compared to Kmeans (7.2%), DPPs selection achieves 9.3% accuracy, validating its ability to reduce redundancy and improve feature coverage. Furthermore, our full method, which integrates DPPs-based selection with HOSVD, achieves the best overall performance at 10.5%, surpassing both previous dataset distillation methods and other alternative sampling techniques. The complete evaluation accuracies are detailed in the supplementary material.

These results highlight the importance of an effective data selection strategy in video dataset distillation. Our approach leverages DPPs to maximize diversity while retaining representative samples, leading to superior generalization in downstream tasks.

| | Evaluation Model | | |
|---|---|---|---|
| | ConvNet3D | CNN+GRU | CNN+LSTM |
| Random | $9.9 \pm 0.8$ | $6.2 \pm 0.8$ | $6.5 \pm 0.3$ |
| DM [45] | $15.3 \pm 1.1$ | $9.9 \pm 0.7$ | $9.2 \pm 0.3$ |
| DM + VDSD [39] | $17.5 \pm 0.1$ | $12.0 \pm 0.7$ | $10.3 \pm 0.2$ |
| MTT [3] | $19.0 \pm 0.1$ | $8.4 \pm 0.5$ | $7.3 \pm 0.4$ |
| MTT + VDSD [39] | $23.3 \pm 0.6$ | $14.8 \pm 0.1$ | $13.4 \pm 0.2$ |
| Ours | $\mathbf{34.8 \pm 0.5}$ | $\mathbf{19.9 \pm 0.7}$ | $\mathbf{18.3 \pm 0.7}$ |

Table 2. Result of experiment on cross-architecture generalization for MiniUCF when IPC is 1.

**Rank Compression Ratio**
We evaluate the impact of different rank compression ratios in HOSVD on overall performance in Tab. 3. Empirical results show that a rank compression ratio of r=0.75 consistently provides a strong balance between storage efficiency and model accuracy across datasets. While increasing the compression ratio reduces storage requirements, overly aggressive compression can lead to significant information loss, negatively affecting downstream tasks. Notably, as shown in Fig. 3, when the rank compression ratio is set to r = 0.1 , both datasets exhibit classification accuracy around 4.0%, suggesting that excessive compression leads to degraded latent representations, making the distilled dataset nearly indistinguishable from random noise.

| Dataset | Rank Compression Ratio | | | | |
|---|---|---|---|---|---|
| | 0.10 | 0.25 | 0.50 | 0.75 | 1.00 |
| MiniUCF | $4.1 \pm 0.1$ | $19.0 \pm 1.3$ | $31.5 \pm 0.7$ | $\mathbf{34.8 \pm 0.5}$ | $28.9 \pm 0.5$ |
| HMDB51 | $3.9 \pm 0.6$ | $7.6 \pm 1.0$ | $11.5 \pm 0.1$ | $\mathbf{12.1 \pm 0.3}$ | $8.9 \pm 0.5$ |

Table 3. Accuracies under different rank compression ratios. Both MiniUCF and HMDB51 datasets are evaluated under IPC 1.

**HOSVD vs Classic SVD** To evaluate the effectiveness of our latent-space compression strategy, we compare truncated SVD with HOSVD under the same storage budget at

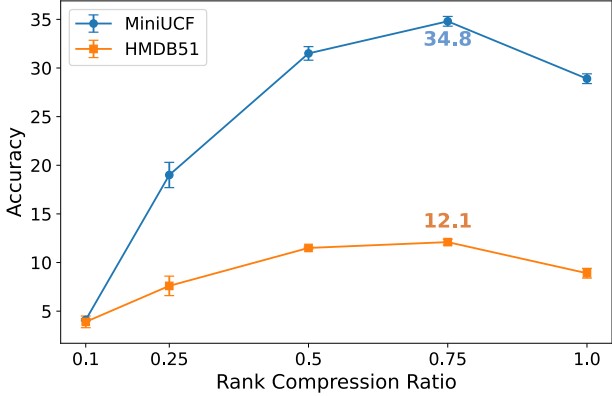

Figure 3. Accuracies of HMDB51 (IPC 1) and MiniUCF (IPC 1) under different rank compression ratios utilized in HOSVD.

IPC 5. Truncated SVD is a matrix factorization technique that approximates a data matrix by keeping only its largest singular values, thereby reducing dimensionality while retaining the most informative components. However, SVD operates on flattened data matrices, leading to a loss of structural information, particularly in spatiotemporal representations.

As shown in Tab. 4, HOSVD consistently outperforms truncated SVD across all datasets, demonstrating its ability to better preserve spatial and temporal dependencies in the latent space. The performance gains are especially notable on MiniUCF (+2.6%) and HMDB51 (+1.8%). Similarly, on Kinetics-400 and SSv2, HOSVD achieves higher classification accuracy (+1.4% and +1.2%, respectively), highlighting its advantage in handling large-scale datasets. These results confirm that HOSVD's tensor-based decomposition provides a more compact yet expressive representation.

| Dataset | MiniUCF | HMDB51 | Kinetics-400 | SSv2 |
|---------|---------|--------|--------------|------|
| SVD | $38.5 \pm 0.4$ | $15.8 \pm 0.2$ | $12.4 \pm 0.3$ | $9.3 \pm 0.2$ |
| HOSVD | $\mathbf{41.1 \pm 0.6}$ | $\mathbf{17.6 \pm 0.4}$ | $\mathbf{13.8 \pm 0.1}$ | $\mathbf{10.5 \pm 0.4}$ |

Table 4. Classification accuracies comparison between different latent compression techniques under the same storage budget for each dataset at IPC 5.

## 4.6. Visualization

Following previous works, we provide an inter-frame contrast between DM and our method to illustrate the differences in temporal consistency in Fig. 4. Specifically, we sample three representative classes (CleanAndJerk, Playing Violin, and Skiing) from the MiniUCF dataset and visualize the temporal evolution of distilled instances. The results clearly demonstrate that our method retains more temporal information, preserving smooth motion transitions across frames. These visualizations further validate the effective-

ness of our latent-space video distillation framework in preserving critical spatiotemporal dynamics.

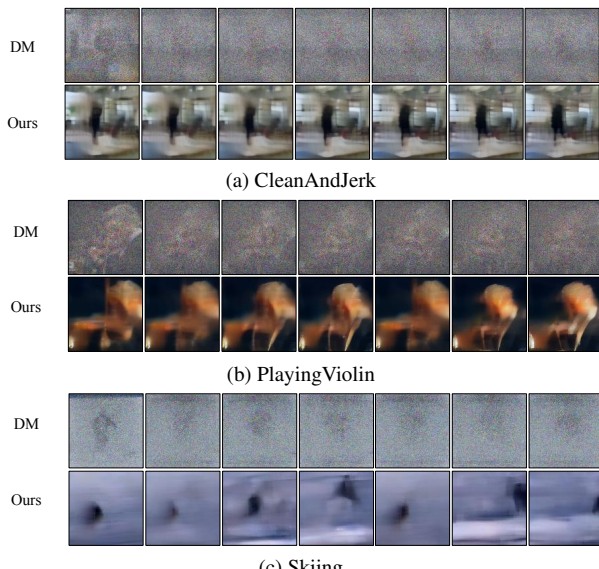

Figure 4. Inter-frame comparison between DM and our method. Our frames are reconstructed from saved tensors and decoded by a 3D-VAE.

## 5. Conclusion

In this work, we introduce a novel latent-space video dataset distillation framework that leverages VAE encoding, Diversity-Aware Data Selection, and High-Order Singular Value Decomposition (HOSVD) to achieve state-of-the-art performance with efficient storage. By applying training-free latent compression, our method preserves essential spatiotemporal dynamics while significantly reducing redundancy. Extensive experiments demonstrate that our approach outperforms prior pixel-space methods across multiple datasets, achieving higher accuracy. We believe our method provides an effective and scalable solution for video dataset distillation, enabling improved efficiency in training deep learning models.

**Future Work** Although we have achieved strong performance using selection-based, training-free methods, there remains room for further improvement. In future work, we aim to explore learning-based approaches in addition to selection-based methods to further enhance dataset distillation performance. By incorporating trainable mechanisms for optimizing distilled representations, we expect to improve both efficiency and generalization. Additionally, we plan to investigate non-linear decomposition techniques for latent-space compression, moving beyond linear factorization methods such as HOSVD.

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
