# Latent Video Dataset Distillation

## Supplementary Material

## 6. VAE

### 6.1. 2D-VAE Quantization

Variational Autoencoders (VAEs) enable significant data compression by encoding each image as a probability distribution in a learned latent space, having the architecture like in Fig. 5. The 2D-VAE used in this paper optimizes the following loss function:

$$\mathcal{L}_{\text{VAE}} = \mathbb{E}_{q_\phi(\mathbf{z}|\mathbf{x})}\left[\log p_\theta(\mathbf{x} \mid \mathbf{z})\right] - D_{\text{KL}}\left(q_\phi(\mathbf{z} \mid \mathbf{x}) \parallel p_\theta(\mathbf{z})\right) \tag{7}$$

The first term minimizes the reconstruction loss when decoding the latent representation of an image, while the second term, the KL divergence, ensures each encoded distribution aligns with a normal prior distribution. Combined, the objective balances the quality of decoded images and the smoothness of the latent distribution.

In order to ensure a fair comparison with previous work, the weights of the VAE are quantized through post-training static quantization, reducing the bid-width from 32 to 8 bits:

$$x_q = \text{round}\left(\frac{x}{s}\right) + z \tag{8}$$

Where $s$ is the scaling factor, and $z$ is the zero point.

By applying linear quantization, the size of the pre-trained model is reduced to one-fourth of its original size. Empirically, the quantized VAE continues to yield high accuracy during experimentation. Compared to other methods such as quantization-aware training, static quantization has the advantage of retaining a high level of accuracy while offering lower computational complexity during the quantization phase.

## 7. Implementation Details

In this section, we provide implementation details of our experiments, including the selection of VAEs, the preprocessing steps applied to video datasets, and the measures taken to ensure a fair comparison.

### 7.1. Additional VAE Selection

We have adopted and quantized SD-VAE-FT-MSE[1] and CV-VAE[47] in our experiments. The variational autoencoders are used to encode video sequences into a compact latent space, enabling efficient dataset distillation. When dealing with IPC 1, where storage constraints are particularly strict, we employ SD-VAE-FT-MSE, a 2D-VAE, which compresses videos as independent frames, allowing

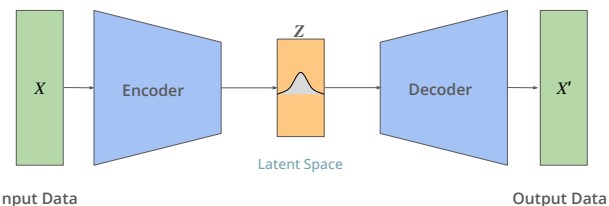

Figure 5. Architecture of Variational Autoencoder(VAE).

for highly compact storage. In contrast, for IPC 5, we utilize CV-VAE, a 3D-VAE, which explicitly models temporal dependencies in video sequences. Unlike 2D-VAEs, which treat frames as separate entities, 3D-VAEs capture motion continuity and temporal redundancy, effectively reducing redundant information across consecutive frames. This results in a more structured latent representation, ensuring that only the most informative motion features are retained, leading to improved efficiency in video dataset distillation. This selective choice of VAE architectures ensures that our distilled datasets achieve the optimal balance between compression efficiency and information retention across different IPC levels.

### 7.2. Quantized VAE Model Size

We apply post-training static quantization on SD-VAE-FT-MSE, compressing the model from original 335MB to 80MB, achieving around 76% compression rate.

### 7.3. Fair Comparison

Throughout our experiments across four video datasets under two IPC settings (1 and 5), we rigorously ensure that the storage used by our method does not exceed predefined storage constraints. For example, in MiniUCF IPC 1, previous methods allocate a storage limit of 115MB. Under the same setting, we sample 24 instances per class and apply HOSVD with a compression rate of 0.75, saving the core tensor and factor matrices. The resulting distilled dataset occupies 27MB, while the quantized 2D-VAE requires 80MB, leading to a total memory consumption of 107MB, which remains within the 115MB storage budget. The detailed storage consumption can be found in Tab. 5.

### 7.4. Sampling Methods

In Tab. 6, we have provided a detailed accuracies on different sampling and dataset distillation techniques evaluating on the dataset SSv2 when IPC is 5.

| Dataset | MiniUCF | HMDB51 | Kinetics-400 | SSv2 |
|---------|---------|--------|--------------|------|
| IPC 1 | 107 MB | 107 MB | 148 MB | 223 MB |
| IPC 5 | 475 MB | 475 MB | 455 MB | 458 MB |

Table 5. Storage consumed by our method for each dataset. Storage represents the total size of the distilled tensors and the associated VAE model.

| Random | DM + VDSD | MTT + VDSD | IDTD | Kmeans | DPPs only | Ours |
|--------|-----------|------------|------|--------|-----------|------|
| $3.9 \pm 0.1$ | $4.0 \pm 0.1$ | $8.3 \pm 0.1$ | $9.5 \pm 0.3$ | $7.2 \pm 0.3$ | $9.3 \pm 0.1$ | $\mathbf{10.5 \pm 0.2}$ |

Table 6. Performance of different dataset distillation and data sampling methods on the SSv2 dataset under IPC 1.

# 8. Peak Memory Analysis

To assess the efficiency of our method in terms of memory consumption, we compare the peak GPU memory usage during dataset distillation with other methods: DM and VDSD. As shown in Tab. 7, our method achieves the lowest peak memory consumption at 11,085 MiB, significantly reducing memory usage compared to DM (20,457 MiB) and VDSD (12,545 MiB).

| Method | DM | VDSD | Ours |
|--------|----|----|------|
| GPU Memory | $20,457$ MiB | $12,545$ MiB | $11,085$ MiB |

Table 7. Peak memory comparsion between different dataset distillation methods on MiniUCF when IPC is 5.

Our method minimizes peak memory usage by operating in the latent space and leveraging training-free compression via HOSVD, significantly reducing redundant memory allocation during dataset distillation. This lower memory footprint allows our approach to scale to larger datasets and higher IPC settings while maintaining efficiency.

# 9. Runtime Analysis

To assess the computational efficiency of our method, we compare its distillation runtime with VDSD across different datasets. All experiments are conducted on an NVIDIA H100 SXM GPU. Our training-free method demonstrates a significant speed advantage, particularly on large-scale datasets, due to its latent-space processing and training-free compression strategy.

On small-scale datasets, such as HMDB51 and MiniUCF, our method completes the dataset distillation process in under 10 minutes, whereas VDSD requires 2.5 hours. The efficiency gain is even more pronounced on large-scale datasets, where our method finishes in approximately 1 hour on Kinetics-400 and SSv2, while VDSD exceeds 5 hours.

These results confirm that our latent-space approach significantly reduces computational overhead compared to pixel-space distillation methods like VDSD. By leveraging structured compression techniques such as HOSVD and

eliminating costly iterative optimization steps, our method achieves faster dataset distillation without compromising performance. This makes our approach highly scalable and practical for real-world applications, especially in large-scale video analysis scenarios.

# 10. Visualization

We provide the reconstructed and decoded frames of our method for MiniUCF across 20 classes in Fig. 6.

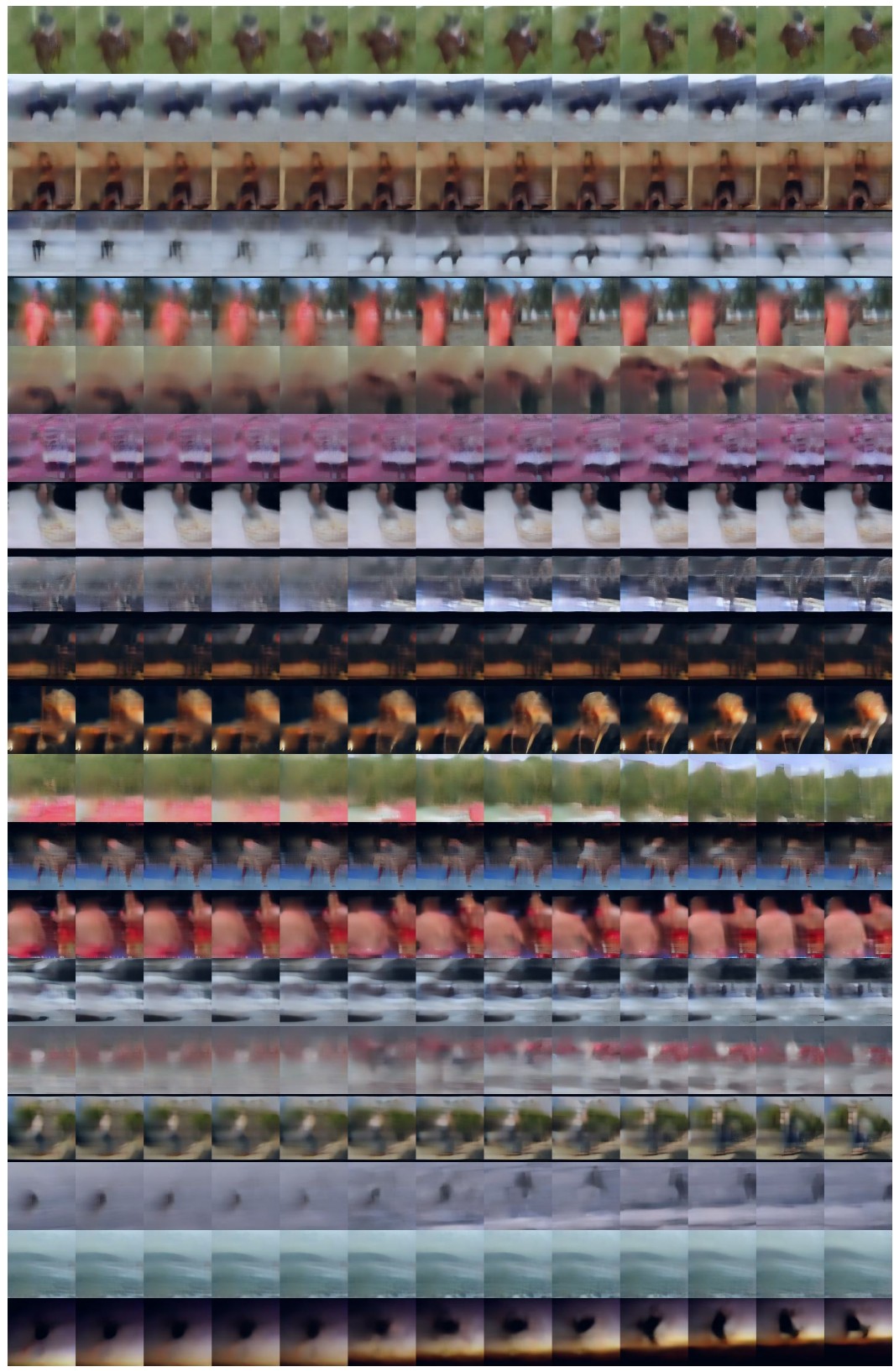

Figure 6. Reconstructed and decoded frames of our method for MiniUCF with a 3D-VAE.