# OpenReview forum: "Latent Video Dataset Distillation"
_thecvf.com/CVPR/2025/Workshop/SyntaGen — SyntaGen 2025 Poster_

### Official Review · Reviewer_MR9w · 2025-03-24

**Rating:** 7
**Confidence:** 3

**Review:**

**Summary**

This paper proposes a novel approach to video dataset distillation, incorporating several key techniques:
1. A Variational Autoencoder (VAE) for efficient encoding of spatiotemporal dynamics.
2. Diversity-Aware Data Selection using Determinantal Point Processes (DPPs) to select a subset of latent embeddings that maximize coverage and informativeness while minimizing redundancy.
3. High-order Singular Value Decomposition (HOSVD), a tensor decomposition method that preserves the original structure of the latent tensor.
4. Quantized VAE, combining dynamic quantization for fully connected layers with mixed-precision optimization for all other layers.

Through extensive experimentation, the proposed method outperforms existing baselines on both small-scale and large-scale datasets.

**Strengths**

This paper may be the first to address video dataset distillation in the latent space, which is more efficient than pixel-based methods by capturing higher-level features. The use of advanced techniques such as DPPs and HOSVD for data selection and compression is innovative. The quantization method reduces model size without sacrificing accuracy, making it particularly appealing for resource-constrained environments. Experimental results demonstrate that the proposed approach outperforms baselines like VDSD and IDTD while requiring less storage.

**Weaknesses**

1. The method's success hinges on the VAE's encoding quality, which may vary with dataset complexity.
2. The method might be sensitive to hyper-parameters, like DPP parameters or HOSVD components.
3. The experiments focus on action recognition, so it's unclear how it performs on tasks like video segmentation or anomaly detection.

---

### Official Review · Reviewer_2aN9 · 2025-03-27
**Evaluation and Reproducibility Concerns**

**Rating:** 6
**Confidence:** 4

**Review:**

The problem setting in this paper is very interesting and well-motivated. However, the explanation of the methodology is presented only at a high level, particularly in the sections on Diversity-Aware Data Selection and the DPPs method, which seem to be the core focus of the paper. Many implementation details crucial for reproducing the method remain unclear, making it difficult to accurately recreate the approach.

Regarding evaluation, the paper appears to demonstrate superior performance over previous methods. However, Figure 2 presents results in a bar chart instead of a table, making it difficult to discern exact numerical values. Additionally, the performance of IDTD in this chart is very high. It even exceeding that of DPP alone. Since the experiment is conducted solely on SSv2, it raises questions about whether IDTD maintains its effectiveness on other datasets or if combining it with the HOSVD step could further enhance performance.

The ablation study primarily examines the effect of different sampling methods but does not explore the impact of variations in SVD. I am particularly curious about whether switching the order—applying HOSVD before DPP—would improve or degrade performance, but the paper does not address this possibility.
 while the distilled dataset demonstrates strong performance in the video classification task, the reconstructed videos suggest that the method may not be well-suited for video generation.

Finally, Despite this limitation, the overall results remain impressive.

---

### Official Review · Reviewer_ToHU · 2025-03-27

**Rating:** 6
**Confidence:** 2

**Review:**

**Summary:**

The paper introduces a new approach to collect latent video dataset, with a focus on reducing the computational and storage challenges. Building upon prior methods that focus on image datasets, the authors propose to:

1. Utilize VAE to encode large video datasets into compact latent space representations.
2. Select diverse data samples with Determinantal Point Processes (DPPs).
3. High-Order Singular Value Decomposition (HOSVD) to compress the latent representations, leveraging the sparsity property of the dataset.

**Strengths:**

- The methodology is well-documented, and the paper provides sufficient experimental validation.
- The empirical results show improvements in video generation quality and efficiency.

**Weaknesses:**

- The paper focuses on video data, but it would be beneficial to discuss the potential applicability of the proposed method to other data modalities, such as audio or multimodal datasets. For example, [1] have explored distillation techniques in the audio domain.  Addressing the adaptability of the proposed approach to different data types could enhance its impact.
- More baselines from other domains such as [1] can also be integrated to improve the experiments.
- While the ablation section provides the effectiveness of each component, more experiments on the contribution of each of the component to the overall success of the method should be provided (e.g. DDP without HOSVD, HOSVD without DDP)

[1] https://web3.arxiv.org/abs/2407.10446

---

### Decision · Program_Chairs · 2025-03-30

**Decision:**

Accept (Oral)

**Comment:**

The paper introduces a frame work for distilling video dataset. Reviewers commented on the novelty and effectiveness and all of them gave positive feedbacks..